# Nationality dominates gender in decision-making in the Dictator and Prisoner's Dilemma Games

Melisa Maya Kumar[1]*, Lily Tsoi[2], Michelle Seungmi Lee[3], Jeremy Cone[4], Katherine McAuliffe[1]

1 Department of Psychology and Neuroscience, Boston College, Chestnut Hill, MA, United States of America, 2 Department of Psychology, Princeton University, Princeton, New Jersey, United States of America, 3 Department of Child Studies and Human Development, Tufts University, Medford, MA, United States of America, 4 Department of Psychology, Williams College, Williamstown, MA, United States of America

* kumarmg@bc.edu

**Data Availability Statement:** All relevant data are within the paper and its Supporting Information files.

**Funding:** The authors received no specific funding for this work.

## Abstract

Across a variety of contexts, adults tend to cooperate more with ingroup members than outgroup members. However, humans belong to multiple social groups simultaneously and we know little about how this cross-categorization affects cooperative decision-making. Nationality and gender are two social categories that are ripe for exploration in this regard: They regularly intersect in the real world and we know that each affects cooperation in isolation. Here we explore two hypotheses concerning the effects of cross-categorization on cooperative decision-making. First, the *additivity hypothesis* (H1), which proposes that the effects of social categories are additive, suggesting that people will be most likely to cooperate with partners who are nationality *and* gender ingroup members. Second, the *category dominance hypothesis* (H2), which proposes that one category will outcompete the other in driving decision-making, suggesting that either nationality or gender information will be privileged in cooperative contexts. Secondarily, we test whether identification with—and implicit bias toward—nationality and gender categories predict decision-making. Indian and US Americans (N = 479), made decisions in two cooperative contexts—the Dictator and Prisoner's Dilemma Games—when paired with partners of all four social categories: Indian women and men, and US American women and men. Nationality exerted a stronger influence than gender: people shared and cooperated more with own-nationality partners and believed that own-nationality partners would be more cooperative. Both identification with—and implicit preferences for—own-nationality, led to more sharing in the Dictator Game. Our findings are most consistent with H2, suggesting that when presented simultaneously, nationality, but not gender, exerts an important influence on cooperative decision-making. Our study highlights the importance of testing cooperation in more realistic intergroup contexts, ones in which multiple social categories are in play.

**Competing interests:** The authors have declared that no competing interests exist.

# Introduction

Cooperation—defined as paying a cost to benefit another individual [1, 2]—yields countless benefits, allowing humans to achieve things together that they could not achieve alone [3, 4]. However, the mutual benefits of cooperation arise only when cooperators work with other cooperators. Otherwise, the efforts of cooperative individuals can be exploited by defectors—those who freeride on the contributions of others [5, 6]. Thus, cooperation in large-scale societies necessitates finding ways to cooperate reliably with strangers [7]. Given the risks associated with working with bad partners, people benefit from carefully selecting their cooperative partners and use strategies to identify partners who are likely to cooperate [1, 8–10]. Group membership, inferred through group markers, serves as an important proxy for identifying potential cooperative partners. When both partners are aware of each other's group membership, levels of cooperation increase [11, 12], indicating that people care about group membership when making cooperative decisions.

Here, we use two salient social categories—nationality and gender, to evaluate how simultaneous membership in these social categories influences cooperative behavior. Specifically, we ask whether levels of cooperation are sensitive to the number of shared social categories. We first review work on ingroup bias in cooperation, demonstrating that social categories influence cooperative behavior when presented in isolation. Next, we review work specifically on cross-categorization, that is, contexts in which more than one social category is used. Finally, we turn to literature on explicit identification and implicit bias which we use as potential predictors of individual variation in cooperative decision-making.

## Ingroup favoritism in cooperation

Many mechanisms have been proposed to explain why we cooperate with our ingroup members more, such as social identity [13], generalized reciprocity [14], obligations [15, 16], and norms [17–19]. These mechanisms are united in broad support of the claim that cooperation is more easily sustained when individuals work with in- as opposed to outgroup members. Indeed, people show ingroup favoritism in their cooperative behavior and this favoritism is seen under both field [20, 21] and laboratory [12, 22] conditions. Economic games used in laboratory experiments are a particularly useful tool for studying group bias in cooperative decision-making because they allow for the controlled manipulation of factors that influence cooperation. In our current study, we investigate whether simultaneously manipulating social categories to which partners belong has an effect on cooperation.

The Dictator Game [23] and the Prisoner's Dilemma [24] have been widely used to study ingroup bias in cooperation. The Dictator Game is a unilateral costly donation paradigm in which player 1, the dictator, decides how much of a windfall endowment to share with player 2, the recipient. It provides a useful measure of sharing and generosity [25–28]. The Prisoner's Dilemma is a two-player game in which the players' decisions to cooperate or defect determine their payoffs. The individual payoff is maximized for the player who defects when the other player cooperates, while mutual cooperation increases partner payoffs, resulting in the highest combined outcome [24]. Both the Dictator Game and the Prisoner's Dilemma are useful measures of cooperation, allowing researchers to explore the factors that influence cooperative decision-making when self-interest is in tension with prosocial (Dictator Game) or mutual (Prisoner's Dilemma) interests (for an example, see [29]). However, unlike the Dictator Game, the Prisoner's Dilemma also serves as a measure of coordination in which players are interdependent to a higher degree [30]. Due to this interdependence, people's decision-making process in the Prisoner's Dilemma includes considering what their partner is going to do—i.e.,

their *beliefs* about whether their partner will cooperate or defect. Overall, players are more likely to cooperate if they believe their partner will also cooperate [31–33].

Group membership influences people's decisions in both the Dictator Game and Prisoner's Dilemma: people share more with ingroup members in the Dictator Game (e.g., [34]) and cooperate more often with their ingroup members in the Prisoner's Dilemma (e.g., [21]). Group bias in the Dictator Game and the Prisoner's Dilemma has been documented with social groups such as nationality [35], religion [35, 36], and ethnicity [36, 37]. People even cooperate more with ingroup members of seemingly arbitrary 'minimal' groups [13, 38–40], showcasing that group membership exerts an important influence on cooperative decision-making. What remains an open question, however, is how more than one social categorization is integrated in cooperative decision-making. In the real world, people are privy to the multiple, salient social categorizations to which potential cooperative partners belong. As reviewed here, people cooperate more with those who share their social categories. Critically, presenting social categories in isolation is not representative of our daily interactions with partners who can be socially categorized via many dimensions. And so, to better understand how social groups influence cooperation, it is essential to observe behavior in contexts in which multiple categorizations are involved simultaneously.

## Nationality and gender as social categories

To examine the effects of two category memberships on cooperative decision-making, it is necessary to select social categorizations that are both salient and shown to be relevant to cooperation. In the present study, we opted to use nationality and gender as social categories of interest because they satisfied these criteria for several reasons. First, with respect to salience, it was important to select social categories that would be easily identifiable through visual markers. People readily encode social group information that is visually salient, such as race, gender and age [41, 42], meaning gender naturally satisfied this criterion. To represent nationality, we used images of flags as a clear visual marker. Second, these social categories inform real-world cooperation: wars are frequently waged based on nation boundaries [43], and division of labor and resources is influenced by gender [44, 45]. Additionally, both nationality and gender have been shown to independently influence people's behavior in economic games [35, 46–48]. Gender has been shown to influence donations in the Dictator Game, yet the directionality of its influence has been mixed in past work. For instance, in one Dictator Game study, both male and female dictators shared more with women [48], while in another, female dictators shared more with men, and male dictators shared similar amounts with women and men [47]. Perhaps more conclusively, in a mega-study with over three-thousand observations, women were more altruistic than men in the Dictator Game, and both men and women expected women to be more altruistic [49]. Nationality has also been shown to influence Dictator Game donations: people share more with own-nationality partners, than with other-nationality partners [35]. Further, in an experimental paradigm with a third-party observing Dictator Game decisions, people punished more often when a dictator was unfair to an own-nationality player [50]. In games with higher interdependence, such as the Prisoner's Dilemma, Balliet and colleagues showed that men tended to be more cooperative than women in same-sex dyads [46]. When playing a similarly interdependent modified Trust Game, Japanese, Chinese and Taiwanese participants trusted their own-nationality partners more than other-nationality partners [12]. Finally, in contrast to social categories in which people choose to belong to (e.g., sports team membership), nationality and gender are both categories to which people tend to belong due to circumstances beyond their control. And so, we expected greater variation in levels of group identification which was important to our secondary question of whether

identification on an explicit and implicit level could explain ingroup biases in cooperative decisions. Of critical importance to the present study is the fact that, in most previous work, nationality and gender categories have been presented in isolation, meaning we do not know how they interact and/or if one category is prioritized over the other. As such, these categories are ripe for the exploration of how multiple group memberships influence cooperative decision-making.

### Effects of crossed categorization

Crossed-categorization refers to the orthogonal combinations of two dichotomized social categories. For instance, crossing nationality (Indian and US American) with gender (female and male) would yield four categorizations: Indian female, Indian male, US American female, US American male. In contrast to dichotomizing group membership on a single category, crossed-categorization is more representative of real-world interactions, where people belong to many social groups simultaneously. Multiple categorization can lead to reduction of intergroup discrimination, but, ingroup bias still persists in these contexts [51–54].

Previous work has emphasized two possible patterns of ingroup favoritism with cross-categorized targets. The first pattern, herein labelled the *additivity hypothesis* (H1), is that ingroup favoritism is highest towards targets that share both social categories (i.e., double-ingroups), lowest towards double-outgroups, and in-between for crossed targets. This prediction assumes that groups will have an additive quality, and is supported by the Category Differentiation Model [55], and Social Identity Theory [56]. For example, when rating cross-categorized targets for attractiveness and competence, nationality and race were additive [57]. That is, participants rated targets that shared their nationality *and* race as being most attractive and competent.

The second possible pattern, herein labelled the *category dominance hypothesis* (H2), is that one social category will dominate the other, and so ingroup bias will vary as a function of whether the dominant category is shared or not. Which social category dominates, depends on the context of the interaction. For example, in one study, participants rated the attractiveness and competence of targets cross-categorized by nationality and race, either following a competitive or cooperative prime [58]. In the competitive prime condition, nationality dominated race: people rated their nationality ingroup members as more attractive and competent than their racial ingroup members. This effect was reversed in the cooperative prime condition, in which racial ingroup members were evaluated as more attractive and competent than nationality ingroup members.

While our primary interest in the present study is to adjudicate between H1 and H2, we recognize that a third possibility is that additivity and category dominance *both* influence decision-making. Consistent with this third possibility, in another crossed-categorization experiment, religious ingroup members were evaluated more positively than nationality ingroup members (i.e., religion dominated nationality), however individuals who shared both categories were evaluated most highly (i.e., religion and nationality were additive; [53]).

Much of the work on cross-categorization has focused on the perception and evaluations of cross-categorized targets. Relatively less work has focused on how cross-categorization influences decision-making. The study that initiated the discussion of cross-categorization in behavioral economics set out to study different aspects of ethnic discrimination in Ashkenazic and Eastern Jews, by measuring behavior across the Trust, Dictator and Ultimatum games [59]. Counter to their predictions, ethnicity-based discrimination was better understood when ethnic and gender identities were considered together. For example, in the Trust Game, transfers to men and women were similar. However, when ethnicity was also considered, transfers

to Ashkenazic women were higher than transfers to Eastern women. Since this study had not intended to study effects of cross-categorization, it does not include analyses on how participant and partner cross-categorizations interact. In other words, while they compare decisions of male participants towards Ashkenazic and Eastern women, they do not consider how Ashkenazic and Eastern male participants would interact with Ashkenazic and Eastern female partners. One study that did explore this influence, used a repeated Prisoner's Dilemma in which players shared *either* religion or ethnicity. They found, consistent with past work, that shared category membership increased cooperation in the Prisoner's Dilemma. However, the addition of a second shared category (i.e., when players shared *both* religion and ethnicity) did not increase cooperation [36]. This study also surveyed the religiosity of their participants, which enhanced the tendency to cooperate more often with religious or ethnic ingroup members, showing the utility of individual-level predictors in helping explain these effects. Our study builds on this work in that we also use a Prisoner's Dilemma to study how cross-categorization affects cooperation. However, we extend this work in important ways. First, we use a one-shot Prisoner's Dilemma, instead of a repeated game, to evaluate cooperative decisions in the absence of reciprocity and reputational concerns which factor into the repeated Prisoner's Dilemma [60]. Second, we examine the effects of cross-categorization in the Dictator Game, a unilateral as opposed to coordinated game, which gives us a measure of cooperation in the absence of coordination.

## Individual-level predictors

In addition to our primary question of how information about partner nationality and gender are integrated in cooperative decisions, we are also interested in exploring two individual-level factors that could be related to these effects: self-reported identification with one's nationality and gender, and implicit evaluations towards the four cross-categorized partners. We were interested in self-reported identification with nationality and gender because one possible explanation as to why one social category might be prioritized over the other could be differential identification with the different social categories. The extent of identification with a social group increases the levels of cooperation with ingroup members [61, 62]. For example, participants who identify with their nationality more than their gender could demonstrate higher rates of cooperation with their nationality ingroup members, over their gender ingroup members. Additionally, we were interested in implicit bias measures because we wanted to assess the relative magnitude of the biases participants show for ingroups based on nationality and gender, and to evaluate whether these implicit biases could explain their allocation decisions. Measures of implicit evaluations assess spontaneous and unintentional evaluations of others and people show ingroup bias on these measures, even when novel minimal groups are used [63, 64].

## Present study

To explore how multiple social categories influence cooperative decision-making, we used a cross-categorization design with nationality and gender. Across two hypothetical economic games, the Dictator Game and the Prisoner's Dilemma, participants were paired with four partners that were in- and outgroup members with respect to both nationality and gender. This fully crossed design allows us to gauge the relative weights of these categories in influencing people's decisions in both unilateral and strategic cooperative contexts.

As outlined above, we were interested in distinguishing between two hypotheses. First, the additivity hypothesis (H1) which proposes that groups will have an additive, compounding quality, with most cooperation towards partners who share both social groups (i.e., double

ingroups), and least cooperation towards partners who don't share either group (i.e., double outgroups), and participants that are ingroups on one dimension falling somewhere in between. Second, the category dominance hypothesis (H2), which proposes that one social category will be prioritized over the other. Specifically, we predict that nationality will override gender in these cooperative contexts, such that people will cooperate more with own-nationality partners than with own-gender partners. We derive this prediction from cross-cultural work that highlights the role of cooperative norms in explaining cross-cultural differences in cooperation [17–19]. In support of this, norms of cooperation are likely transmitted within a culture [37, 65, 66]. Consequently, when cooperating with own-nationality partners, participants can form beliefs about how prospective partners are likely to behave. Hence, in the context of the Dictator Game and Prisoner's Dilemma, we predict that decisions might rely on culturally-bounded norms, yielding higher rates of cooperation with own-nationality partners than with own-gender partners. We also explored the possibility that individual variation in identification with these social categories, and implicit evaluations of partners may relate to the patterns of cooperation.

Understanding how ingroup favoritism in cooperation is influenced by multiple social categories is essential, since it is more representative of interactions with real-world partners. In the present study, we observe people's cooperative decisions in this rich context of cross-categorization, where multiple social categories are at play.

## Materials and methods

This study has been approved by the Institutional Review Board at Boston College, under protocol number 16.209.01. All participants gave their written consent prior to the study.

### Participants

Indian and US American Amazon Mechanical Turk workers (*N* = 479) were tested (69 Indian Females, 104 Indian Males, 129 US American Females and 177 US American Males). For our dataset, please see S1 Data in S1 File collection was done in three waves, between September 2017 and August 2018. The first of the three waves slightly differed from the other two in their comprehension check questions and the Affect Misattribution Procedure, the task we used to measure implicit bias (for more details on the recruitment and the data cleaning procedure please see S2 Appendix in S1 File). An additional 187 participants were tested but excluded because they failed comprehension checks either for the Dictator Game (4.65% of the total) or the Prisoner's Dilemma (18.47% of the total), or reported speaking Mandarin or Cantonese (12.61% of the total), which was an exclusion criterion for the Affect Misattribution Procedure. Participants reported which age, salary and education brackets they fell into (a breakdown of this demographic information can be found in the S1 File, see S3 Table in S1 File). Note that our sample size is imbalanced with respect to the four social categorical combinations due to the availability of mTurk workers, as well as a differential propensity to pass comprehension checks and the language criteria.

### Design

The study consisted of four measures: (1) identification questions, (2) the Dictator Game, (3) the Prisoner's Dilemma, and (4) the Affect Misattribution Procedure. The identification questions were always presented first, and the Affect Misattribution Procedure last, preceding demographic questions. The order of the two hypothetical economic games was randomized. The Dictator Game and Prisoner's Dilemma both included comprehension questions to ensure that the participants understood the rules of the game. Performance on these questions

served as a basis for exclusion. The surveys used in the study are available as (see S4 Appendix in S1 File).

## Materials and procedure

**Identification questions.** Participants were asked their nationality and gender, along with the following questions: "How important is your nationality/gender in describing who you are?" and "To what extent do you identify with your nationality/gender?" Participants responded to these four questions on a 10-point scale, anchored by 'not important at all' and 'extremely important', and 'very little' and 'a great deal', respectively. The order in which participants responded to the nationality and gender questions was randomized. Post-hoc analyses revealed a strong positive correlation between the two nationality questions, ($r$(475) = .89, $p < .001$), and the two gender questions, ($r$(474) = .72, $p < .001$). In subsequent analyses, responses to these questions were averaged to obtain two scores, one reflecting identification with nationality and the other with gender.

**Dictator Game (DG).** Participants were told they would complete a four-round task as Player A. In each round they would be playing with a different Player B, and could give any amount of the $10.00 endowment they receive at the beginning of each round, to Player B. It was made clear that all decisions and partners were hypothetical in the beginning of the game (for surveys, see S4 Appendix in S1 File).

Following the instructions, participants were asked one comprehension check question depicting a possible decision (i.e., to transfer $3.00 to Player B) and asking how much would be left for Player A. If answered correctly (i.e., $7.00), participants would move on to play the DG. If answered incorrectly, participants were given another chance to respond. If the second attempt failed as well, a third opportunity to respond was accompanied with the instructions to the game. Participants who failed to respond correctly after three attempts (4.65%) were excluded.

Participants then played DGs with the following Player B's, presented in randomized order: Indian female, Indian male, US American female and US American male. All Player B's were accompanied by avatars indicating nationality and gender, through hair length and a flag (see Fig 1A). Each round, participants were presented with eleven multiple choice options, ranging from 'Transfer $0.00 to Player B (keep $10.00 for myself)' to 'Transfer $10.00 to Player B (keep $00.00 for myself), with increments of $1.00.

**Prisoner's Dilemma (PD) decisions and beliefs.** Participants were told they would complete a four-round task as Player A. In each round they would be playing with a different Player B, and both players would have the choice to cooperate or defect by transferring or keeping their endowment of $2.00 per round. All possible outcomes were spelled out and illustrated with a PD matrix (see Fig 1B). It was made clear that all decisions and partners were hypothetical at the beginning of the game.

Following the instructions, participants were asked four comprehension check questions depicting all four possible decision pairs, and asking how much the payoff for Player A would be if a given decision were made. If all four questions were answered correctly, participants moved on to play the PD. If answered incorrectly, participants were given another chance to respond. If the second attempt failed as well, a third opportunity to respond was accompanied by the instructions to the game. Participants who failed to respond correctly on all three attempts for any of the four questions were excluded (18.47%).

Participants played the PD with the following Player B's, presented in randomized order: Indian female, Indian male, US American female and US American male. As in the DG, all Player B's were accompanied by avatars indicating nationality and gender, through hair length

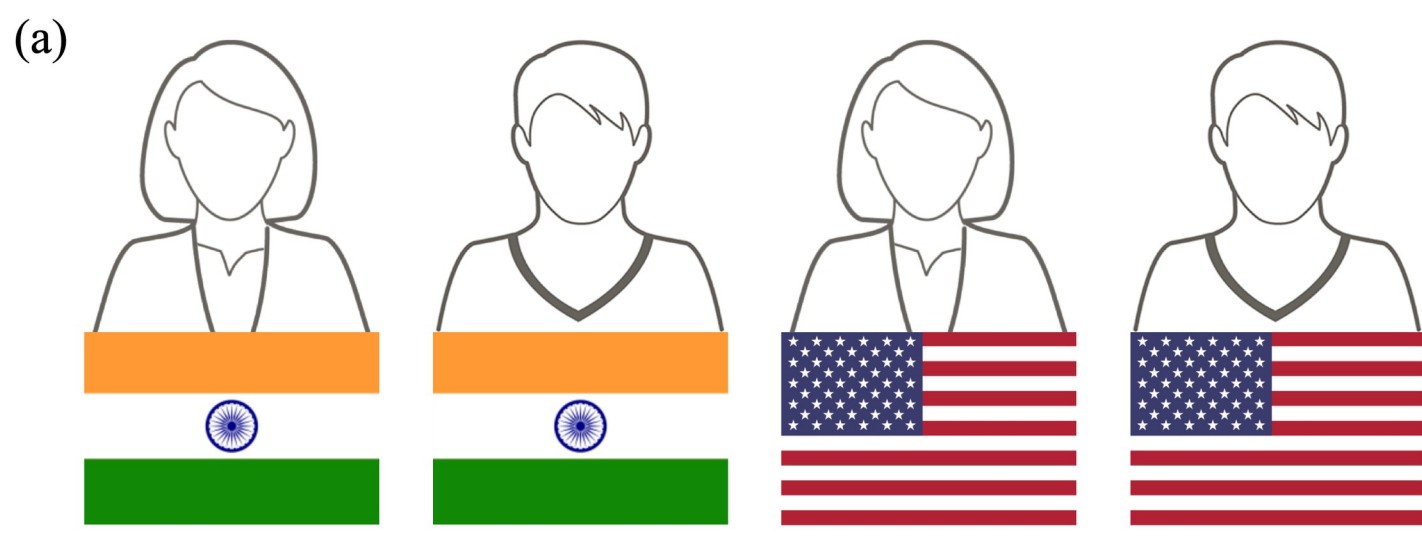

**Fig 1. Stimuli used in the study.** (a) Avatars that represented (from left to right) Indian female, Indian male, US American female, US American male partners in the Dictator Game and Prisoner's Dilemma, as well as presented during the Affect Misattribution Procedure. (b) The Prisoner's Dilemma payoff matrix that participants saw as instructions. The participant, Player A, had two options: they could transfer $2.00 to Player B or keep $2.00 for themselves. Payoffs on the left (italicized) are participant payoffs, while payoffs on the right are Player B payoffs.

and a flag (see Fig 1A). Each round, participants were asked about both their own decision to cooperate or defect (i.e., transfer or keep), as well as what they thought their partner would choose to do (i.e., transfer or keep). People's decisions to cooperate or defect are henceforth referred to as PD decisions, while their beliefs about what their partner would do are referred to as PD beliefs.

**Affect Misattribution Procedure (AMP).** To measure participants' implicit evaluations towards each target group, they completed an Affect Misattribution Procedure (AMP; [67]). The AMP is a sequential priming task in which participants are required to evaluate a neutral stimulus that is presented in quick succession after a prime over multiple trials. Specifically, each of the 120 trials in the measure consisted of: (a) a prime image (75 ms) of the avatars used in the economic games (30 trials for each social category, see Fig 1A), (b) an inter-stimulus interval (ISI; 125 ms), (c) a Chinese pictograph (100 ms), and, (d) a backward mask (white noise), which remained on the screen until participants provided a response [67]. Participants'

task was to indicate whether the Chinese pictograph was more or less pleasant than average by using two keys, 'k' for more pleasant, and 'd' for less pleasant. Payne and colleagues [67] have demonstrated that participants' judgments about the Chinese pictographs are influenced by the primes that occur before they are presented, even when participants are asked to avoid having their judgments influenced by the primes. Thus, performance on the AMP can be used as an indirect measure of participants' spontaneous and unintentional evaluations of the primes.

AMP scores were the proportion of pleasant judgments, ranging from 0 (i.e., no pleasant ratings at all) to 1 (i.e., all pleasant ratings) for each of the four types of targets: Indian females, Indian males, US American females and US American males. From these four scores, we calculated two measures of the extent to which participants exhibited an implicit preference for each of the social categories: one for nationality (i.e., American primes vs. Indian primes) and one for gender (i.e., female primes vs. male primes). To this end, we calculated difference scores by subtracting the average score for the outgroup, from the average score for the ingroup. For example, for a US American female participant, the AMP nationality difference score would be the difference between her average AMP score for her nationality ingroup members (i.e., average of her AMP score for US American female and her AMP score for US American male), and her average AMP score for her nationality outgroup members (i.e., average of her AMP score for Indian female and her AMP score for Indian male). Each of these two types of AMP difference scores, reflecting implicit ingroup biases, was then entered as separate predictors of participants' choices in each of the economic games.

Since the Chinese pictographs were used to capture the pleasantness ratings towards the social categories, it was important that our participants did not encode these pictographs as meaningful. To control for this, participants who reported speaking Mandarin or Cantonese were excluded (12.61%).

## Coding and analysis

Responses to all components of our task, except for the AMP, were recorded on Qualtrics [68]. The AMP was hosted on a separate web server. Analyses were conducted in R version 3.6.3 [69]. Models were run using the package 'lme4' [70].

We were primarily interested in examining how simultaneous social group membership influenced behavior in the DG and PD. Our dependent variables (DVs) were (a) donations out of $10.00 in the DG (continuous DV, ranging from 0 to 10 with increments of 1), (b) decisions to cooperate or defect in the PD (binary DV, cooperate coded as 1, defect coded as 0), and (c) beliefs about whether partners would cooperate or defect in the PD (binary DV, belief partner will cooperate coded as 1, belief partner will defect coded as 0). DG behavior was analyzed using Linear Mixed-Effects Models (LMM), with a continuous dependent variable. PD behavior was analyzed using Generalized Linear Mixed-Effects Models (GLMM). In all models, we included participant identity as a random intercept to control for repeated measures within participants. We report F-tests for LMM analyses, and likelihood ratio test (LRT) results for GLMMs.

In our Results section that follows, we first report the results of identification and implicit bias as a manipulation check, to assess whether our participants indeed cared about nationality and gender categories. To test our additivity hypothesis (H1)—i.e., whether nationality and gender were additive in terms of their influence on cooperation—we first coded partner social category membership, from the perspective of the participant. This yielded two 2-level factors: nationality group status (ingroup or outgroup) and gender group status (ingroup or outgroup). For example, to a US American female participant, an Indian female partner would be a nationality outgroup and gender ingroup member. This coding scheme allowed us to look at

whether shared nationality, shared gender or shared nationality *and* gender were significant predictors of levels of cooperation across our three main DVs. We ran separate models predicting each of our three DV's (DG donations, PD decisions and PD beliefs) with the two-way interaction term, nationality group status X gender group status.

To test our category dominance hypothesis (H2)—i.e., whether one social category dominated the other in terms of its influence on cooperative decision-making—we examined whether nationality or gender was prioritized in DG donations, PD decisions, and PD beliefs. We used two interaction terms as predictors in our models: participant nationality with partner nationality, and participant gender with partner gender. If, for instance, shared nationality is more predictive of cooperative behavior than shared gender (i.e., if H2 is confirmed such that nationality dominates gender, the direction we predict), the nationality interaction will be a significant predictor across DVs, while the gender interaction will not.

For both the additivity (H1) and category dominance (H2) analyses, we ran the same models including the following covariates as a robustness check: age, salary, education, and wave of data collection. The pattern of results is unchanged. Results from these models can be found in S5 (additivity analysis) and S6 (category dominance analysis) Files in S1 File.

Finally, we examined whether the patterns of bias can be explained by our two individual-level predictors: (1) identification with nationality and gender and (2) implicit pleasantness judgments as measured by the AMP. We ran twelve separate models here, varying the DV (i.e., DG donations, PD decisions, or PD beliefs), predictor of interest (identification or AMP), and dimension of categorization (nationality or gender). For identification, we had two scores per participant: nationality identification and gender identification. In our models, we used three-way interaction terms (for nationality: participant nationality X partner nationality X nationality identification, for gender: participant gender X partner gender X gender identification) to predict each of our DVs. For the AMP, we also had two scores per participant: a nationality difference score and a gender difference score (detailed in the AMP methods section). We used these scores in place of the identification variables in the three-way interactions described above (e.g., participant nationality X partner nationality X AMP nationality difference score).

## Results

### Preferences for nationality and gender: Identification & implicit bias

For the most part, participants in our study cared about their nationality and gender, both in terms of explicit identification and on an implicit evaluative measure, as captured by the AMP (see Fig 2). Participants in our study reported identifying with both their nationality ($M$ = 7.27 on a ten-point scale, $SD$ = 2.75), and gender ($M$ = 8.08, $SD$ = 2.17. Independent samples *t*-tests revealed that Indians identified more with their nationality ($M$ = 8.73, $SD$ = 1.8) than US Americans did ($M$ = 6.45, $SD$ = 2.85; $t(471.08)$ = 10.71, $p < .001$, $d$ = 0.9) and females identified more with their gender ($M$ = 8.33, $SD$ = 1.94) than males did ($M$ = 7.9, $SD$ = 2.31;, $t(462.87)$ = 2.19, $p$ = .03, $d$ = 0.2).

Participants also showed an implicit preference for nationality ingroup members over the nationality outgroup members ($t(478)$ = 8.54, $p < .001$, $d$ = 0.39). Parallel to identification, Indians showed more implicit bias towards their own-nationality ($M$ = 0.16, $SD$ = 0.33), than US Americans did ($M$ = 0.09, $SD$ = 0.27; $t(296.11)$ = 2.22, $p < .05$, $d$ = 0.22. Female participants showed an implicit preference for their own gender ($M$ = 0.03, $SD$ = 0.11). Male participants, on the other hand, showed a slight implicit preference for females ($M$ = -0.05, $SD$ = 0.17).

### Were nationality and gender additive (H1)?

Nationality and gender were not additive in the DG, PD decisions, or PD beliefs. Participants did not favor double ingroup partners over partners that shared either category, or double

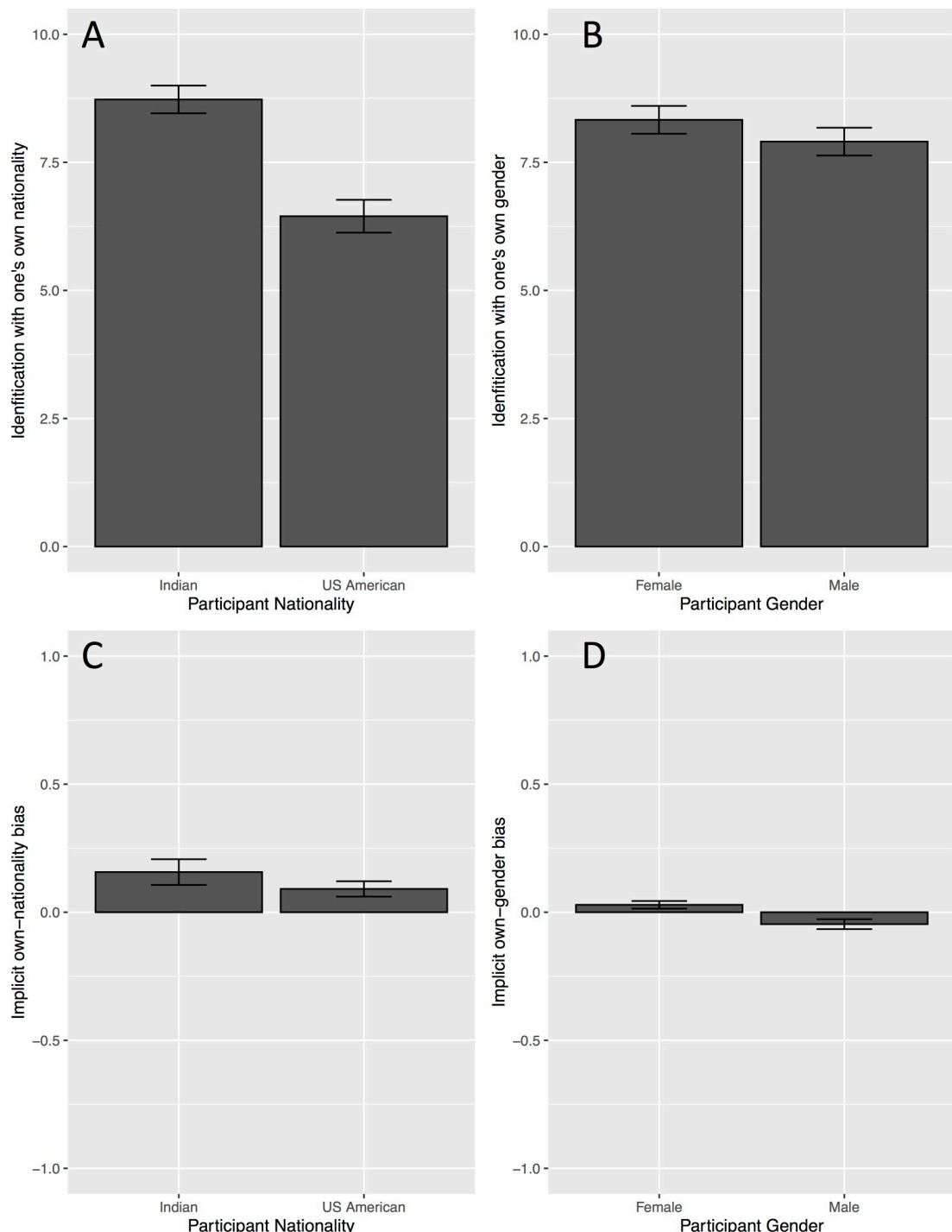

**Fig 2. Participants explicitly identified with both their nationality and gender, and implicitly preferred their nationality.**
Females also have an implicit own-gender bias, while males have a female bias. (a) Nationality identification by participant
nationality on the left, (b) gender identification by participant gender on the right. (c) Nationality difference scores in the AMP
plotted against participant nationality (left), (d) and gender difference scores in the AMP plotted against participant gender
(right). AMP difference scores (c, d) above 0 indicate ingroup bias in our implicit measure, while scores below 0 indicate
outgroup bias. All error bars represent 95% confidence intervals.

outgroup partners. None of our interaction terms (Nationality group status X Gender group status) significantly predicted our DVs (all $p$'s > .18), meaning cooperative behavior did not increase as a function of how many social categories were shared. For the model output table and figure of this analysis, please see S4 Appendix in S1 File. Next, we turn to the question of whether nationality or gender was the dominant social category that explains bias in cooperative decisions.

### Did one social category dominate the other (H2)?

Overall, participants cooperated more with own-nationality partners than other-nationality partners (Fig 3). Across the three DVs, our model including interactions between (a) participant nationality X partner nationality and (b) participant gender X partner gender showed that the two-way interaction between participant and partner nationality was a significant predictor of cooperative behavior (DG: F(1, 1432) = 96.37, $p$ < .001, PD decisions: LRT, $\chi^2(1)$ = 19.96, $p$ < .001, PD beliefs: LRT, $\chi^2(1)$ = 29.32, $p$ < .001; see Table 1 for model outputs). In the DG, Indians gave more to Indians; and US Americans gave more to US Americans. In the PD, participants chose to cooperate more with own-nationality partners. Participants' beliefs matched their decisions: they expected more cooperation from own-nationality partners. Indians were overall more 'optimistic' in their PD behavior, cooperating and expecting cooperation at a higher rate than US American participants.

The participant gender X partner gender interaction was not a significant predictor of cooperation in our three DVs (DG: F(1, 1432.1) = 1.32, $p$ = .25, PD decisions: LRT, $\chi^2(1)$ = 2.4, $p$ = .12, PD beliefs: LRT, $\chi^2(1)$ = 2.8, $p$ = .09). Further, participant gender was not predictive of cooperation in any of the measures. However, partner gender was a significant predictor of behavior in all DVs (DG: F(1, 1433.06) = 5.77, β = -.1, $SE$ = .04, $p$ = .016, PD decisions: LRT, $\chi^2(1)$ = 12.77, β = .57, $SE$ = .16, $p$ < .001, PD beliefs: LRT, $\chi^2(1)$ = 19.64, β = .63, $SE$ = .15, $p$ < .001). Participants shared and cooperated more with female partners than with male partners, and believed that female partners would cooperate more often than male partners.

### Do individual-level predictors explain these findings?

Having established that nationality dominated gender, here we focus on whether identification to nationality or implicit evaluations of own- and other-nationality partners can help explain these effects. Model output tables for identification and the AMP, figures for the AMP, as well as our gender analyses not explained here, can be found in S7 File in S1 File. Our three-way interaction term, participant nationality X partner nationality X nationality identification, was only predictive in the DG (F(1, 1431.03) = 7.47, β = 0.12, $SE$ = .04, $p$ = .006). The extent to which participants identified with their nationality only mattered in the DG. Specifically, participants who identified more with their nationality, shared slightly more with own-nationality partners, and this was particularly pronounced in Indian participants. However, the extent to which participants identified with their nationality was negatively related to sharing overall.

AMP nationality difference scores were predictive of behavior across all our measures. For the DG donations, PD decisions and PD beliefs, the three-way interaction participant nationality X partner nationality X AMP nationality difference score was significant (DG: F(1, 1431.03) = 89.84, β = 2.66, $SE$ = .28, $p$ < .001, PD decisions: LRT, $\chi^2(1)$ = 36.99, β = -6.96, $SE$ = 1.28, $p$ < .001, PD beliefs: LRT, $\chi^2(1)$ = 53.11, β = -8.01, $SE$ = 1.26, $p$ < .001). Consistent with the predictions, in the DG, the greater the difference between a participants' implicit preference for their own-nationality over the other-nationality, the more participants shared with their own-nationality partners (S7 Fig, top row in S1 File). By contrast, in the PD, this pattern of results was reversed: Indian and US American participants were more likely to cooperate

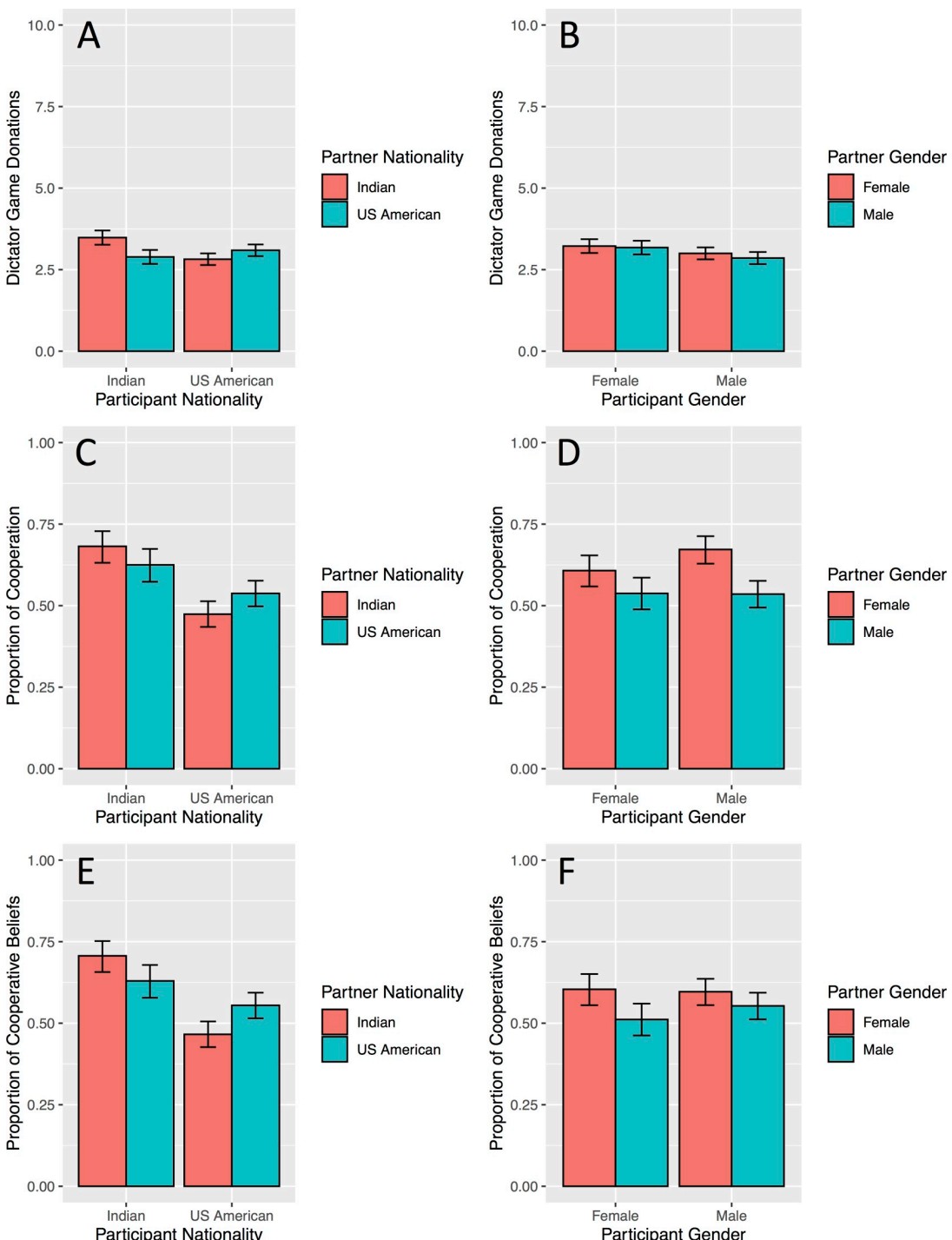

**Fig 3. Participant and partner nationality is overall more predictive of cooperation than participant and partner gender.**
Donations in the Dictator Game (a, b), the proportions of decisions to cooperate in the Prisoner's Dilemma (c, d) and the proportions of beliefs that partners will cooperate in the Prisoner's Dilemma (e, f) shown by nationality (left column) and gender (right column). All error bars represent 95% confidence intervals.

with their own-nationality partners when their implicit preference was towards other-nationality partners (S7 Fig, middle row in S1 File). Additionally, beliefs of partners cooperating mapped onto decision making (S7 Fig, bottom row in S1 File). That is, the more participants

**Table 1. Models for category dominance across the Dictator Game, Prisoner's Dilemma Decisions, and Prisoner's Dilemma Beliefs.**

|  | Dictator Game | Prisoner's Dilemma Decisions | Prisoner's Dilemma Beliefs |
|---|---|---|---|
| (Intercept) | 3.67 (0.20)*** | -3.38 (0.60)*** | -2.63 (0.43)*** |
| Participant nationality | -0.67 (0.20)*** | 3.04 (0.59)*** | 2.58 (0.43)*** |
| Partner nationality | -0.59 (0.07)*** | 0.63 (0.25)* | 0.73 (0.24)** |
| Participant gender | -0.23 (0.19) | 0.69 (0.53) | 0.13 (0.39) |
| Partner gender | -0.04 (0.07) | 0.87 (0.26)*** | 0.92 (0.23)*** |
| Participant nationality X Partner nationality | 0.86 (0.09)*** | -1.46 (0.33)*** | -1.61 (0.30)*** |
| Participant gender X Partner gender | -0.10 (0.09) | -0.51 (0.33) | -0.49 (0.29) |
| AIC | 6580.82 | 1772.70 | 1966.10 |
| BIC | 6630.84 | 1817.17 | 2010.55 |
| Log Likelihood | -3281.41 | -878.35 | -975.05 |
| Number of trials | 1915 | 1916 | 1913 |
| Number of participants | 479 | 479 | 479 |
| Variance: Participant ID (Intercept) | 3.87 | 18.82 | 10.41 |
| Variance: Residual | 0.86 | | |

*Note.* AIC = Akaike information criterion; BIC = Bayesian information criterion; Num. obs. = number of observations; Var: subject variance associated with participant id (random intercept term). Baselines are as follows: US American for both participant and target nationality; and male for both participant and target gender.

***$p < 0.001$

**$p < 0.01$

*$p < 0.05$.

implicitly preferred their own-nationality, the more likely they were to believe that other-nationality partners would cooperate.

## Discussion

### Summary

This study investigated the effects of multiple social categories on cooperative decision-making. We sought to adjudicate between two hypotheses: (1) *the additivity hypothesis* (H1), which proposes that the influence of social categories will be additive such that we will see highest rates of cooperation when partners share *both* nationality and gender and (2) *the category dominance hypothesis* (H2), which proposes that one social category will dominate the other in driving decision-making such that either nationality or gender information will be prioritized in cooperative contexts. Our data are most consistent with the category dominance hypothesis (H2). Specifically, when making cooperative decisions, nationality is privileged over gender. In the DG, people shared most with nationality ingroup members. In the PD, people cooperated more with nationality ingroup members, and believed that their nationality ingroup partners would cooperate with them. By contrast, shared gender was not as strong of a predictor of sharing, cooperation or expectations of cooperation. The extent to which participants identified with their nationality influenced sharing in the DG: high-identifiers shared less overall but more with their own-nationality partners. In this same vein, an implicit preference for own-nationality partners resulted in more sharing in the DG. Surprisingly, however, implicit own-nationality preferences were associated with less cooperation and beliefs about cooperation in the PD.

### Our two hypotheses

This study brings data to bear on two hypotheses regarding the effects of cross-categorization on cooperative decision-making. One hypothesis is that nationality and gender would be

additive (H1): people would cooperate most with double ingroups, least with double out-groups, and that partners sharing one social category would be in between. Second, we hypothesized that if either dimension of social categorization was found to be dominant, it would be nationality (H2). Our findings support H2: sharing in the DG, decisions to cooperate, and beliefs that the partner will cooperate in the PD, were higher when paired with an own-nationality partner. By contrast, there was no consistent gender ingroup bias across these measures. Note that our study was designed to understand how cooperation is influenced by simultaneous categorization. And so, these findings do not speak against the possibility that gender effects may emerge when gender is presented in isolation.

One explanation for why nationality appears to be prioritized over gender in the context of cooperative decision-making is that nationality could provide a more relevant boundary within which cooperative norms exist and are transmitted. Cooperative behavior in economic games shows marked variation at the cross-societal level [17–19, 65], which has largely been attributed to differences in economic organization and social structure. Perhaps shared nationality provides a template of normative information (e.g., Indians help one another) over and above what gender can provide. In support of this idea, some work suggests that gender differences observed in the DG can be explained by nationally transmitted gender norms [71, 72]. Gong and colleagues [71] found that women in a matrilineal ethnic minority group, the Mosuo, shared less than men, and there were no gender differences in the neighboring patriarchal group, the Yi. This finding suggests that gender-specific norms of cooperation are overridden by the overarching ethnic norms. In a similar vein, the 'women are wonderful' effect, such that women are seen as more positive and caring than men [73], is smaller in more gender-egalitarian societies [74, 75], supporting the idea that gender stereotypes are moderated by culture [76]. These explanations lend nicely to our findings that while there are some effects of partner gender, participant behavior is bounded more strictly by nationality.

Although additivity reliably emerges in cross-categorized partner evaluations (for a review see [77]), nationality and gender were not additive in our study. In other words, in the context of cooperative decision-making, sharing both nationality and gender did not lead to highest levels of cooperation. Indeed, relative to nationality, gender played a minor role in informing decisions. However, gender did inform one context: in the PD people cooperated more with female partners and believed that female partners would cooperate more often. It is noteworthy that this partner gender effect was not present in the unilateral DG, but was in the coordinated PD. In contrast to the DG, in the PD, mentalizing—the ability to reason about what the partner might do–becomes more important [78]. Why did a pro-female bias emerge in the PD where people consider what their partner would do? One possibility is that the previously mentioned 'women are wonderful' effect is at play: This global evaluation of women as positive and caring may lead their partners believing that women will cooperate, and so deciding to cooperate with them in the PD. In experimental support of this idea, both women and men expect women to share more in the Dictator Game [49]. This expectation may have motivated both men and women to cooperate with women at higher rates. Another possibility is that women's stronger ingroup bias underlies this asymmetry [79], and so while women showed ingroup bias in the PD, men did not. However, it is also possible that men were particularly motivated to cooperate with women: perhaps men seized an opportunity to signal their cooperativeness to women [80, 81], an explanation that would be broadly consistent with sexual selection theory [82]. Although it was not the case in the cooperative contexts of the DG and PD, gender could still be additive in different contexts, or when cross-categorized with a different social category, such as age. Future work should explore these possibilities.

## Individual-level predictors

Given the category dominance of nationality over gender, we focus our individual differences section on whether identification with one's own nationality, and implicit evaluations of own-nationality over other-nationality partners can help explain the patterns of cooperation. Identification with nationality, and implicit preferences for own-nationality over other-nationality partners, predicted more sharing with own-nationality partners in the DG. The more participants identified with their nationality, the more they donated to nationality ingroup over nationality outgroup partners. This is in line with work by Ando [61], who also found that people who identified more with their social group cooperated more with them. A possible mechanism for this may be social projection: in their meta-analysis, Robbins and Krueger [62] proposed social projection (i.e., I am similar to others), as a key factor when making judgments about others. Individuals are more likely to engage in social projection with ingroup members, which could explain the downstream consequence of higher rates of cooperation. Additionally, implicit preferences for own-nationality over other-nationality partners, also led to more sharing with own-nationality partners in the DG.

While identification with nationality did not influence cooperative decisions and beliefs in the PD, implicit evaluations of own- over other-nationality partners did. Unexpectedly, implicitly evaluating own-nationality partners as better, resulted in less cooperation with own-nationality partners in the PD. Beliefs of partner cooperation mapped onto these decisions: participants who implicitly preferred their own-nationality believed that other-nationality partners would cooperate more often. For instance, Indian participants who implicitly preferred Indians, were more likely to cooperate with US Americans. Perhaps these measures influenced DG donations in the predicted direction, but not the PD, because donations in the DG can be interpreted as a measure of preference. Higher identification and higher implicit bias, is similarly indicative of a preference towards own- over other-nationality partners. In turn, we see coherence in how identification and implicit preference influences the DG. The PD, however, involves strategic coordination, and so the partner is integral to the outcome of the game. Our design used hypothetical partners, while the AMP measures real implicit bias towards members of these social groups. Perhaps implicit bias did not explain PD in the predicted direction, because of this asymmetry in 'realness'. That is, mentalizing about hypothetical partners in the PD, may have led to different behaviors and beliefs than what people's real implicit biases would predict. Still, it is unclear whether this could account for an effect in the opposite direction—where higher own-nationality implicit bias was associated with less cooperation with own-nationality partners. In the PD, implicit own-nationality biases manifested as 'optimism' towards what other-nationality partners would do. Perhaps to continue to reap the benefits of cooperation with other-nationality partners, people with high implicit own-nationality biases assumed that partners who do not share their social category would share their preference for this category (e.g., Indian participants would expect US Americans to positively evaluate Indian partners). This could help explain the optimism surrounding other-nationality partner behavior: i.e., people might have believed that other-nationality partners would cooperate and so decided to cooperate with them as well. Since partners become more relevant in the PD compared to the DG, future work could manipulate the levels of identification and levels of implicit bias of the *partners* in the PD, to see how these factors come into play in games with higher interdependence. Although speculative at this stage, we view this as an intriguing possible explanation for this unexpected pattern of findings, one that warrants future exploration.

### Limitations and future directions

One potential limitation of this study is that in the administration of the AMP, the salience of nationality as a categorization might have been higher than that of gender, since the flag is a stronger visual cue (see Fig 1). However, previous work has shown that implicit bias on the AMP is unaffected by manipulations of attention to different ways of categorizing others [83]. Specifically, Gawronski and Ye [83] conducted an AMP procedure where participants implicitly evaluated Black or White men that were old or young. Participants were told to pay attention to either race, or age; and the attentional manipulation was not significant. That is, participants' AMP scores were not affected by whether they were paying attention to one social categorical dimension over the other. Even if our participants paid more attention to the nationality cue over the gender cue, it is unlikely that evaluations in the AMP were due to attention in our task.

Another potential limitation of this study is that the decisions and partners were hypothetical. This is not a huge concern since previous work has shown comparable decisions across hypothetical and incentivized games [27, 84–87], but also see [88]. Indeed, levels of cooperation in our DG (M = 30.4%, SD = 2.19) are comparable to that of incentivized iterations of the game. For example, in their meta-analysis of Dictator Games, Engel (2011) [27] reported contributions of 28.3%. In other work, contributions of 43.8% were reported for economic games where payoffs were hypothetical, and so did not depend on decisions in the game (i.e., no-stakes games), and 33.2% for incentivized games [86]. Despite these comparisons, it stands to reason that levels of cooperation may change slightly with real incentives and partners. However, the hypothetical nature of our tasks does not bear on our main question of whether nationality and gender were additive or one was dominant over the other. That is, although absolute levels of cooperation may differ in situations with real partners, we see no reason to expect that the dominance of nationality over gender would change in an incentivized task.

Finally, we chose nationality and gender as our social categories since these categories are salient, provide significant variation in identification, and have shown to influence cooperative decision-making. However, to achieve a fuller understanding of how cross-categorization may influence cooperation, it will be important to extend the present questions to other social groups.

## Conclusion

In the context of the two social categories that we examined—nationality and gender—we provide evidence for category dominance (H2), and not additivity (H1). Indians and US Americans shared more and cooperated more often with own-nationality partners, regardless of their gender. In other words, nationality represents a more significant boundary on cooperation than does gender. This study marks an important contribution to work on cross-categorization because it focuses specifically on how cross-categorization influences cooperative decision-making. It not only examines decision-making in both unilateral and coordinated games, but also explores whether implicit evaluations towards crossed-categorized partners influence cooperation. In doing so, this study brings us one step closer to understanding cooperation in the real world—where people belong to multiple social categories simultaneously.

## Supporting information

**S1 File.**
(ZIP)

## Author Contributions

**Conceptualization:** Melisa Maya Kumar, Lily Tsoi, Michelle Seungmi Lee, Jeremy Cone, Katherine McAuliffe.

**Data curation:** Melisa Maya Kumar, Lily Tsoi.

**Formal analysis:** Melisa Maya Kumar, Lily Tsoi, Jeremy Cone, Katherine McAuliffe.

**Investigation:** Melisa Maya Kumar, Michelle Seungmi Lee, Jeremy Cone, Katherine McAuliffe.

**Methodology:** Melisa Maya Kumar, Michelle Seungmi Lee, Jeremy Cone, Katherine McAuliffe.

**Project administration:** Melisa Maya Kumar, Katherine McAuliffe.

**Resources:** Jeremy Cone, Katherine McAuliffe.

**Software:** Melisa Maya Kumar, Jeremy Cone.

**Supervision:** Lily Tsoi, Jeremy Cone, Katherine McAuliffe.

**Visualization:** Melisa Maya Kumar, Katherine McAuliffe.

**Writing – original draft:** Melisa Maya Kumar.

**Writing – review & editing:** Melisa Maya Kumar, Lily Tsoi, Michelle Seungmi Lee, Jeremy Cone, Katherine McAuliffe.

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
