## [Decision Letter · Decision Letter 0]

4 Sep 2020

PONE-D-20-20922

Nationality dominates gender in decision-making across two cooperative games

PLOS ONE

Dear Dr. Kumar,

Thank you for submitting your manuscript to PLOS ONE. After careful consideration, we feel that it has merit but does not fully meet PLOS ONE’s publication criteria as it currently stands. Therefore, we invite you to submit a revised version of the manuscript that addresses the points raised during the review process.

Please find below the reviewer's comments, as well as those of mine.

We look forward to receiving your revised manuscript.

Kind regards,

Valerio Capraro

Academic Editor

PLOS ONE

Additional Editor Comments:

I have now collected one review from one expert in the field. The reviewer likes the paper, but has one major reservation, concerning the hypothetical nature of the experiment. I have read the paper and I share the reviewer's point. Therefore, I would like to invite you to revise your work following the reviewer's comments and paying particular attention to the issue above. Moreover, I would like to note that I have a published work showing that women are expected to be more altruistic than men (Branas-Garza et al. 2018). It seems to me that this work is very related to your work.

I am looking forward for the revision.

References

Brañas-Garza, P., Capraro, V., & Rascon-Ramirez, E. (2018). Gender differences in altruism on Mechanical Turk: Expectations and actual behaviour. Economics Letters, 170, 19-23.

Journal Requirements:

2. Please modify the title to ensure that it is meeting PLOS’ guidelines (https://journals.plos.org/plosone/s/submission-guidelines#loc-title). In particular, the title should be "specific, descriptive, concise, and comprehensible to readers outside the field" and in this case it is not informative and specific about your study's scope and methodology.

Reviewers' comments:

Reviewer's Responses to Questions

**Comments to the Author**

1. Is the manuscript technically sound, and do the data support the conclusions?

Reviewer #1: Partly

2. Has the statistical analysis been performed appropriately and rigorously? 

Reviewer #1: Yes

3. Have the authors made all data underlying the findings in their manuscript fully available?

Reviewer #1: Yes

4. Is the manuscript presented in an intelligible fashion and written in standard English?

Reviewer #1: Yes

5. Review Comments to the Author

Reviewer #1: The paper deals with an important topic, the interaction of different social categories and their effect on social behavior. It builds on a large literature on identification with and judgement based on social categories, and adds to this literature that it looks at cross categorization and its effects on behavior, not just on judgement.

It uses two economic games, a dictator game and a prisoner´s dilemma game, to look at contexts with a) one-shot interactions and b) pure preference vs. coordinating interaction.

The authors use nationality and gender as the two social categories and implement this in an M-Turk study, using flags and stylized faces with long or short hair as indication of the categories. The design is within-subject, such that each participant plays with a double ingroup, ingroup-outgroup/outgroup-ingroup and double outgroup partner in random order, for both games.

The authors find a dominant category, nationality, and not an additive effect of the two categories. In addition to the game, they use an implicit bias task to estimate implicit bias of each participant for both categories and test its influence on behavior. Here, they find somewhat confusing results, as implicit bias leads to more cooperative behavior towards the ingroup in the dictator game and less cooperative behavior towards the ingroup in the prisoner´s dilemma.

Expectations about partner´s behavior are also measured and do not have an effect in the nationality perspective, but in the gender perspective women are expected to be more cooperative.

Overall, the question is important, the paper is well written and the results seem mostly clear, with the exeption of the implicit attitude analyses.

I see one major problem with the design of the study: The authors use hypothetical scenarios, where participants know that the partners do not exist. In such a setting, neither contributions to a partner nor expectations about the partner make much sense, as they know that there are no partners.

This shows most clearly in the results of the implicit bias task, where implicit bias seems to be counterintuitively linked to the behavior in the game. The authors propose some arguments for that, but leave out the argument that it could just be the hypothetical character of the partners driving that - while the implicit bias is real, the behavior is not really oriented towards a "partner". This could also explain the difference between the dg and the pg - in a pg, the partner and its characteristics are more important, as the authors rightly state, and if it is clear that there is no "real" partner, it has a larger effect than in a setting where partner and characteristics are less important as compared to own preferences.

The fact that payment is not based on results in the game will probably further this problem - it has been shown in the literature that in many games there are no large differences whether played hypothetically or for small amounts of money, but usually pro-sociality is a bit overstated in hypothetical games.

Finally, the authors miss one of the fundamental papers that started the discussion of cross-categorization in experimental economics, Fershtmann & Gneezy 2001.

6. PLOS authors have the option to publish the peer review history of their article (what does this mean?). If published, this will include your full peer review and any attached files.

Reviewer #1: No

---

## [Author Response · Author response to Decision Letter 0]

23 Nov 2020

Dear Dr. Capraro,

Thank you for the positive evaluation of our manuscript, and encouragement to submit a revision. We have addressed the concerns you and Reviewer 1 have raised, and our manuscript has been improved substantially as a result of your thoughtful input. Most notably, we have focused on further explaining the hypothetical nature of our design by highlighting our main between-condition comparisons and extending our discussion to more clearly address how this aspect of our design may have affected our results. We have also, as suggested by Reviewer 1, further clarified the methods and analyses of our implicit attitude task. Finally, we have incorporated the literature suggested by you and Reviewer 1. 

We believe that our manuscript is now stronger thanks to the revisions detailed below and would make a significant contribution to PloS1. We hope you agree and look forward to your feedback. 

EDITOR COMMENTS

E1. I have now collected one review from one expert in the field. The reviewer likes the paper, but has one major reservation, concerning the hypothetical nature of the experiment. I have read the paper and I share the reviewer's point. Therefore, I would like to invite you to revise your work following the reviewer's comments and paying particular attention to the issue above. 

Thank you for your thoughtful comments on our paper. We address your and Reviewer 1’s major concern below in response R.2.

E2. Moreover, I would like to note that I have a published work showing that women are expected to be more altruistic than men (Branas-Garza et al. 2018). It seems to me that this work is very related to your work.

Thank you for bringing this work to our attention, we have incorporated it into our introduction and discussion on pages 6 and 26, respectively. This paper is indeed very relevant to our work and has strengthened our argument about gender-based expectations in cooperation. 

REVIEWER COMMENTS

R1. Overall, the question is important, the paper is well written and the results seem mostly clear, with the exception of the implicit attitude analyses.

Thank you for your helpful feedback on our manuscript and your suggestions on how to improve it. We address each of your comments in turn below, including your comments on the implicit attitude analyses. We agree that the implicit attitude (AMP) analyses were not explained as clearly as they should have been in our original submission. For instance, in our original submission, the AMP data were explained in the Coding and Analysis section, as opposed to in the Methods section when the measure was first introduced. We have now moved this section on the AMP to an earlier section of the paper under the Methods, on page 16. In this section, we have also provided further details about the AMP measure, and how the nationality and gender difference scores were calculated from the AMP data. This hopefully helps readers understand the AMP data and analyses better, prior to coming to the results section. Additionally, we have extended our explanation of the AMP analyses in the results, and referenced the relevant figures that can be found in the supplement, on pages 23 and 24. 

R2. I see one major problem with the design of the study: The authors use hypothetical scenarios, where participants know that the partners do not exist. In such a setting, neither contributions to a partner nor expectations about the partner make much sense, as they know that there are no partners. 

We agree that the hypothetical nature of this task represents a potentially important limitation to our study, and one that we now address in more detail in our revised discussion. That being said, we would like to note that previous work investigating economic decision-making in tasks including the DG and PDG have found relatively few differences between incentivized and hypothetical instantiations of the tasks. We now elaborate on these past studies in our revised discussions on pages 28-30. 

One way to address the concern that rates of cooperation in an incentivized version of our task may differ from the hypothetical version is to compare directly the rates that we found to those reported in the literature. For example, in their metaanalysis of Dictator Games, Engel et al (2011) found contributions of 28.3% andour participants contributed 30.4% (SD = 2.19) on average. Another study directly compared games with no-stakes to incentivized games (Amir, Rand & Gall, 2012). For the DG, they report contributions of 43.8% in the no-stakes condition, and 33.2% in the incentivized condition. Taken together, these studies suggest that our DG contributions (i.e., 30.4%) are at least broadly consistent with those in incentivized games. Amir et al. (2012) also reported comparisons for the incentivized vs. hypothetical versions of theUltimatum Game (UG). The UG is arguably somewhat similar to the PD as they are both games that call upon beliefs about the partner’s behavior. Here, real stakes were associated with higher offers (49.7% versus 46.1% in no-stakes). However, another meta-analysis on the UG finds no effect of stake size on offers (Larney, Rotella & Barclay, 2019). Rates of cooperation in our PD ranged from 47.4% (US American participants with Indian partners) to 68.2% (Indian participants with Indian partners), with an average rate of 58%. The higher rates that we find may be due to differences in the UG and PD, or to the overestimation of cooperation in our hypothetical game. We raise these ideas in our revised discussion.

Nevertheless, we agree that absolute levels of cooperation may vary across hypothetical and incentivized tasks, and perhaps especially in the PD where the partner is integral to the outcome of the interaction. Having said that, our task was principally designed to assess between-partner differences in cooperation as opposed to absolute levels. Specifically, we do not expect that an incentivized version of these games would change the overall pattern of our results with respect to nationality dominating gender in these contexts. Indeed, in our revised manuscript (e.g., p. 29-30), we have been careful not to draw strong inferences about the rates of cooperation across our participants and rather have restricted our claims to relative levels of cooperation as measured by DG and PD depending on partner type and participant characteristics. 

R3. This shows most clearly in the results of the implicit bias task, where implicit bias seems to be counterintuitively linked to the behavior in the game. The authors propose some arguments for that, but leave out the argument that it could just be the hypothetical character of the partners driving that - while the implicit bias is real, the behavior is not really oriented towards a "partner". This could also explain the difference between the dg and the pg - in a pg, the partner and its characteristics are more important, as the authors rightly state, and if it is clear that there is no "real" partner, it has a larger effect than in a setting where partner and characteristics are less important as compared to own preferences.

Thank you for raising this point. Indeed, our results showing that implicit attitudes do not consistently map on to behavior in the PD, may partly be due to the hypothetical partners. We have added this possible explanation to our discussion, on page 28. We were also puzzled by the AMP results and agree that the asymmetry in ‘realness’ with the AMP and PD may contribute to these findings. Still, it is unclear how this would explain the effect in the opposite direction (i.e., implicit own-nationality preferences predicted more cooperation with other-nationality partners) that we find, a point we also raise in our revised discussion. 

R4. The fact that payment is not based on results in the game will probably further this problem - it has been shown in the literature that in many games there are no large differences whether played hypothetically or for small amounts of money, but usually pro-sociality is a bit overstated in hypothetical games.

We agree that prosociality may be overestimated in hypothetical situations. Based on our comparison of contributions in incentivized vs hypothetical DGs as detailed in R2, we believe our rates of cooperation are broadly consistent with those reported in incentivized tasks, particularly with respect to the DG. However, we agree that rates of cooperation that we saw in the PD may be slightly inflated, potentially due to our hypothetical design— a limitation we now acknowledge in our manuscript. As mentioned above (R2) we have been careful in our discussion to restrict our claims to relative and not absolute rates of cooperation. Specifically, we suggest that although levels of cooperation may decrease in an incentivized games, it is unlikely that our main effect that nationality dominated gender (and that nationality and gender were not additive) would change. 

R5. Finally, the authors miss one of the fundamental papers that started the discussion of cross-categorization in experimental economics, Fershtmann & Gneezy 2001.

Thank you for pointing out this article, we have incorporated it into our introduction on pages 8-9.

---

## [Decision Letter · Decision Letter 1]

14 Dec 2020

Nationality dominates gender in decision-making in the Dictator and Prisoner’s Dilemma games

PONE-D-20-20922R1

Dear Dr. Kumar,

We’re pleased to inform you that your manuscript has been judged scientifically suitable for publication and will be formally accepted for publication once it meets all outstanding technical requirements.

Kind regards,

Valerio Capraro

Academic Editor

PLOS ONE

Additional Editor Comments (optional):

Reviewers' comments:

Reviewer's Responses to Questions

**Comments to the Author**

1. If the authors have adequately addressed your comments raised in a previous round of review and you feel that this manuscript is now acceptable for publication, you may indicate that here to bypass the “Comments to the Author” section, enter your conflict of interest statement in the “Confidential to Editor” section, and submit your "Accept" recommendation.

Reviewer #1: All comments have been addressed

2. Is the manuscript technically sound, and do the data support the conclusions?

Reviewer #1: Yes

3. Has the statistical analysis been performed appropriately and rigorously? 

Reviewer #1: Yes

4. Have the authors made all data underlying the findings in their manuscript fully available?

Reviewer #1: Yes

5. Is the manuscript presented in an intelligible fashion and written in standard English?

Reviewer #1: Yes

6. Review Comments to the Author

Reviewer #1: (No Response)

7. PLOS authors have the option to publish the peer review history of their article (what does this mean?). If published, this will include your full peer review and any attached files.

Reviewer #1: No

---

## [Editor Report · Acceptance letter]

30 Dec 2020

PONE-D-20-20922R1 

Nationality dominates gender in decision-making in the Dictator and Prisoner’s Dilemma Games 

Dear Dr. Kumar:

I'm pleased to inform you that your manuscript has been deemed suitable for publication in PLOS ONE. Congratulations! Your manuscript is now with our production department. 

Kind regards, 

on behalf of

Dr. Valerio Capraro 

Academic Editor

PLOS ONE